# Measures to assess quality of postnatal care: A scoping review

**Anna Galle**[1]*, **Allisyn C. Moran**[2], **Mercedes Bonet**[3], **Katriona Graham**[4], **Moise Muzigaba**[2], **Anayda Portela**[2], **Louise Tina Day**[5], **Godwin Kwaku Tuabu**[4], **Bianca De Sá é Silva**[4], **Ann-Beth Moller**[3]

1 Department of Public Health and Primary Care, WHO Collaborating Centre on Primary Care and Family Medicine, University Centre for Nursing and Midwifery, Ghent University, Belgium, 2 World Health Organization Department of Maternal, Newborn, Child and Adolescent Health and Ageing, Geneva, Switzerland, 3 World Health Organization Department of Sexual and Reproductive Health and Research, Development and Research Training in Human Reproduction (HRP), UNDP/UNFPA/ UNICEF/WHO/World Bank Special Programme of Research, Geneva, Switzerland, 4 Department of Public Health and Primary Care, WHO Collaborating Centre on Primary Care and Family Medicine, Ghent University, Belgium, 5 Department of Infectious Disease Epidemiology, Maternal Newborn Health Group, London School of Hygiene & Tropical Medicine, London, United Kingdom

* anna.galle@ugent.be

**Data Availability Statement:** All data and related metadata underlying the findings reported in the paper are available in the Supporting Information files.

## Abstract

High quality postnatal care is key for the health and wellbeing of women after childbirth and their newborns. In 2022, the World Health Organization (WHO) published global recommendations on maternal and newborn care for a positive postnatal care experience in a new WHO PNC guideline. Evidence regarding appropriate measures to monitor implementation of postnatal care (PNC) according to the WHO PNC guideline is lacking. This scoping review aims to document the measures used to assess the quality of postnatal care and their validity. The review was conducted according to the Preferred Reporting Items for Systematic reviews and Meta-Analyses extension for Scoping Reviews (PRISMA-ScR). Five electronic bibliographic databases were searched together with a grey literature search. Two reviewers independently screened and appraised identified articles. All data on PNC measures were extracted and mapped to the 2022 WHO PNC recommendations according to three categories: i) maternal care, ii) newborn care, iii) health system and health promotion interventions. We identified 62 studies providing measures aligning with the WHO PNC recommendations. For most PNC recommendations there were measures available and the highest number of recommendations were found for breastfeeding and the assessment of the newborn. No measures were found for recommendations related to sedentary behavior, criteria to be assessed before discharge, retention of staff in rural areas and use of digital communication. Measure validity assessment was described in 24 studies (39%), but methods were not standardized. Our review highlights a gap in existing PNC measures for several recommendations in the WHO PNC guideline. Assessment of the validity of PNC measures was limited. Consensus on how the quality of PNC should be measured is needed, involving a selection of priority measures and the development of new measures as appropriate.

**Funding:** This work received support from the Bill and Melinda Gates Foundation. AG, AM and ABM received salary from the funders. All other authors did not receive any financial provision for their contribution from the funders. The funders had no role in the study design, data collection and analysis, decision to publish, or preparation of the manuscript.

**Competing interests:** The authors have declared that no competing interests exist.

## Introduction

The days and weeks following childbirth – the postnatal period – is a critical time for women and newborns [1]. Major physical, social and emotional changes occur during this period, yet this is often the most neglected period on the continuum of maternal and newborn care, challenged by the fragmentation of services [2]. Quality postnatal care (PNC) services can have a lifelong positive impact on health and well-being of women, newborns, and children, facilitating a supportive environment for the parents, caregivers and families [3–6].

In March 2022, the World Health Organization (WHO) published a new guideline entitled "WHO recommendations on maternal and newborn care for a positive postnatal experience", to improve the quality of postnatal care – including provision and experience – for women and newborns with the ultimate goal of improving their health and well-being. Within the guideline, a positive postnatal experience is defined as "an experience in which women, newborns, partners, parents, caregivers and families receive information, reassurance and support in a consistent manner from motivated health workers; where a resourced and flexible health system recognizes the needs of women and babies and respects their cultural context" [7]. The WHO PNC recommendations in the guideline address clinical and non-clinical maternal and newborn care, health promotion and health systems interventions during the six-week period (42 days) after birth [1]. It updates and expands upon the 2014 PNC recommendations and complements WHO existing guidelines on the management of postnatal complications [7]. The recommendations have a wide scope, are for all settings and include aspects such as early child development and mother-infant bonding.The 2022 WHO PNC guideline proposes that implementation and impact of these recommendations should be monitored at facility, subnational and national levels. Monitoring should be based on clearly defined criteria and with measures that are associated with locally agreed targets.

WHO has placed provision and experience of care are at the core of the WHO framework for improving the quality of care for mothers and newborns around the time of childbirth, including the postnatal period [8]. Within this framework, outcomes include coverage of key practices and people centred outcomes. Currently, tracking PNC coverage at global level is recommended by two contact indicators: proportion of women or newborns receiving PNC care in the health facility or at home in the first 48 hours after childbirth [9–11]. There are also widely used measures that assess the content of postnatal care for the mother and newborn e.g., the percentage of women who received breastfeeding support, immunization of the newborn and provision of postpartum family planning services [12,13]. In low-and middle -income settings, these measures are typically collected from population-based household surveys (e.g., the Demographic and Health Surveys (DHS) database and the Multiple Indicator Cluster Surveys (MICS) [14,15] and some from routine health information systems. Only recently, maternal and newborn health experts have emphasized the need to measure not only coverage and content of PNC, but to also include quality measures focusing on how interventions are delivered by including people centred outcomes [2,16]. Despite a growing number of measures to assess quality of PNC in the literature [17,18], globally agreed standards are lacking [19,20], and there are limitations in terms of validity.

Recent reviews have focused on antenatal care measures and measures on the experience of facility-based care for pregnant women and newborns, but no similar exercise has been conducted regarding PNC [21,22]. Reviews and guidance documents on broader maternal and newborn health indicators are available as well [23–25], but often focus is on facility-based care in the first 24 hours which overlooks the 6-week postnatal period and especially care for the woman and newborn at home. This scoping review aims to document the availability and validity of existing PNC measures for quality of postnatal care described in the peer-reviewed and grey literature based on the 2022 WHO PNC guideline.

## Materials and methods

### Study design

The study protocol was registered in Open Science Framework [26] (reference 10.17605/OSF. IO/PSFXB) and outlines the methodology for the design and conduct of the scoping review. The review process was based on the Arksey and O'Malley's five step scoping review framework: i) identifying the research question; ii) identifying relevant studies; iii) selecting studies; iv) charting the data; and v) collating summarizing and reporting the results [27]. Results are presented according to the Preferred Reporting Items for Systematic reviews and Meta-Analyses extension for Scoping Reviews (PRISMA-ScR) Checklist (http://www.prisma-statement. org/Extensions/ScopingReviews) [28] (**S1 Checklist**).

### Definitions

"Measures" are used by governments, researchers, and other institutions to assess the quality of PNC, including provision and experience of care. In this review measures can be broadly described as the level or state of an object under study, whereas indicators more narrowly are considered as indirect representations with a quantitative nature [29,30]. By using measures as an umbrella term, all kind of assessments of quality of care will be considered in this review, including those informed by qualitative research. PNC is defined as care of women after childbirth and newborns, including the promotion of healthy practices, disease prevention, and detection and management of problems during the first six weeks after birth [1].

### Inclusion and exclusion criteria

The PCC (Population/Concept/Context) framework of the Joanna Briggs Institute (JBI) [31] was used to articulate the search strategy, inclusion and exclusion criteria.

Inclusion criteria:

- Peer reviewed scientific and grey literature, published between 2010 and 2022.

- Quantitative, qualitative and mixed methods studies.

- Describing quality of PNC, including provision or experience of care.

- -ndividual studies or systematic reviews focused on developing or improving indicators related to PNC.

- Studies in languages mastered by the review team: English, Spanish, Portuguese, French, German, and Arabic.

Exclusion criteria:

- Studies focusing solely on medical health outcomes, for example in the context of randomized controlled trials new therapies.

- Systematic reviews not including the development or improvement of indicators, to avoid double inclusion of studies (as a single study and in a review).

- Studies only reporting on PNC uptake (Yes/No) in one indicator, without including any measurement of provision or experience of care. Reporting on the actual content (=provision) or experience of PNC was a requirement for inclusion.

- Publications without detailed methods and findings (e.g., conference abstracts, study protocol, commentary).

## Search strategy

The scoping review included literature searches from 1st January 2010 to 1st March 2022. Search strategies were developed for PubMed, Embase, Scopus, Web of Science and CINAHL databases (S1 Text). Grey literature was obtained using the Google search engine by reviewing the first ten pages. In addition, the websites of the Demographic and Health Surveys (DHS) database, the Multiple Indicator Cluster Surveys (MICS), the Reproductive Health Survey (RHS) database, the United States Agency for International Development (USAID) publications and WHO publications [14,15,32–34] were consulted. Lastly, the Global Index Medicus was reviewed for both grey literature and peer reviewed articles. Keywords for these searches were postnatal care, quality of care and measures as this database does only support use of simple keywords. Search terms concerning quality of care were developed based on the definition of quality of care of both the Institute of Medicine (IOM) and WHO frameworks [35,36]. The IOM established six domains of health care quality in 2001, which have been re-utilized by many other stakeholders, including WHO. The six dimensions state that health care should be safe, effective, patient-centered, timely, efficient, equitable. The WHO framework of quality of care (2018) defines eight domains of quality of care, with quality standards for each domain, and encompasses both the provision and experience of care [37].

## Study selection

Citations were imported into Rayyan, an online tool developed to support the screening and data extraction processes [23]. Two independent reviewers (AG and GT) screened the title and abstract, followed by full-text assessment of potentially eligible studies. Conflicts were resolved upon discussion with a third reviewer (ABM).

## Data extraction and analysis

A data extraction form in Microsoft Excel was piloted with the first 15 articles and adapted with input from ABM, AG, BS and GTK. The final data extraction table included authors, publication year, purpose of study, year of study, study design, geographical location of study by country and WHO region [38], measures used for assessing PNC, if measures were clearly defined, if any indicator testing or validation performed (such as reliability, accuracy, feasibility), funding sources and declared conflicting interests. Data-extraction was conducted by a team of three reviewers (AG, BS and GTK). BS and GTK divided the articles for data extraction, while AG reviewed all articles and extracted all the data independently as a double check. The completed data extraction sheet can be found as a supplementary file (S1 Table).

The measures were then mapped according to the 2022 WHO PNC recommendations [7], first by the three guideline categories: i) maternal care; ii) newborn care; and iii) health system and health promotion interventions and then subsequently to specific recommendations within these categories (Fig 1). Experience of care measures identified (e.g., did you feel respected?) were mostly not related to specific PNC recommendations and therefore where only mapped according to the three overarching guideline categories. In this manuscript we refer to the studies (including scientific studies and grey literature reports) with identified measures, while the supplementary file (S1 Table) gives a an overview of all identified measures per study. In addition, a narrative description was given of the content of the measures. The results were further analyzed using descriptive numerical summary analysis and narrative synthesis.

To give an overview of what is available and what is missing in the current literature regarding quality of care measures for PNC according to the 2022 WHO PNC guideline, reasons for gaps in the literature were analyzed according to the framework of Robinson et al. (2011) [39]. The framework proposes 4 categories of reasons for research gaps: i) insufficient or imprecise

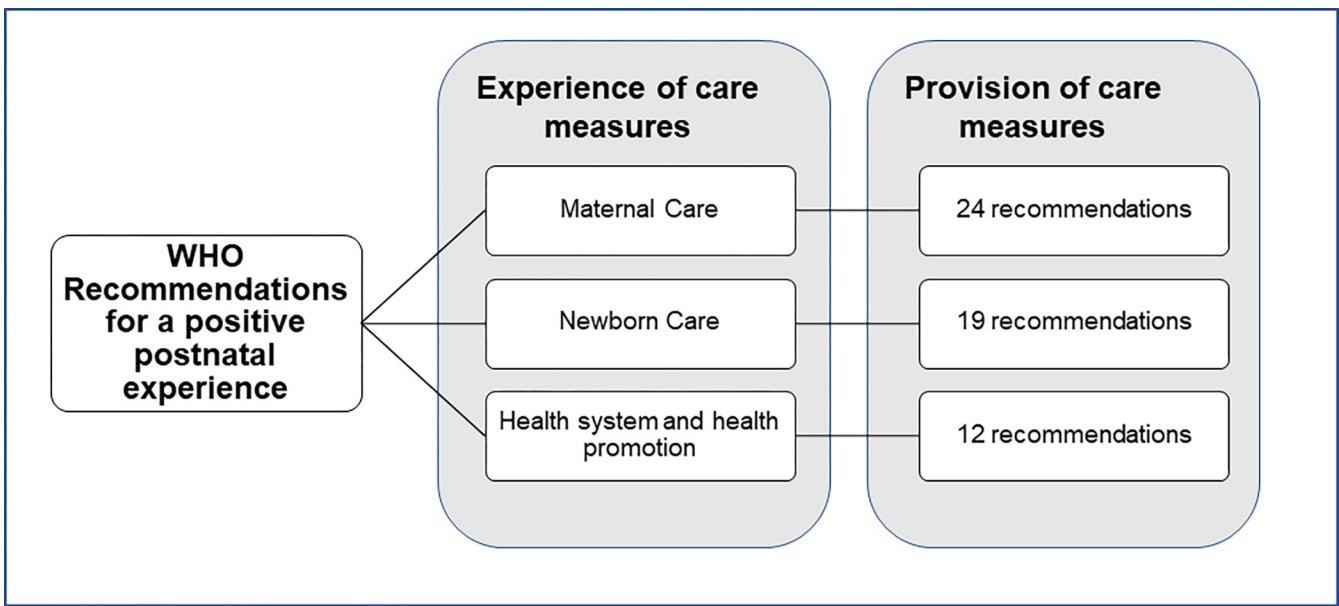

**Fig 1. Mapping framework for PNC measures.**

information, ii) biased information, iii) inconsistent results, iv) not the right information. All reasons for research gaps were described narratively within the given categories, only the last category was not applicable to this review.

## Results

The bibliographic database searches initially identified 2743 titles and abstracts. The grey literature search identified additional 31 titles and abstracts. After removal of duplicates, 2017 title and abstract were screened, resulting in 127 potential records to be included. After full text screening, 62 records (48 peer reviewed articles and 14 grey literature documents) were included in the review. A flowchart based on PRISMA guidelines is presented in Fig 2 [40].

### Study characteristics

Studies most commonly took place in the African Region (n=20; 32%), followed by the European Region (n=10; 16%) and the South-East Region (n=10; 16%) (Table 1). Five studies (8%) took place in the Region of the Americas and five studies (8%) in the Western Pacific Region. Only one (3%) took place in the Eastern Mediterranean Region and 11 (18%) in countries from multiple regions. The most used study design was a cross sectional design (n=38, 61%) followed by a (quasi-) experimental design for evaluating interventions (n=5, 8%). Five included studies (8%) were national or international guideline documents on PNC services and six (10%) were validation studies. Another eight studies (13%) aimed at developing a new scale or instrument. Most included studies had clear definitions of their measurements, but six out of 62 articles (10%) did not have clear definitions for the used measures or definitions were not clear. Most studies (n=19, 31%) relied on data collected from women surveys followed by data extracted from population-based household data (n=11, 18%). Seven studies (11%) relied on data collected from observations and four studies (7%) used health facility records as data source. Thirteen studies (21%) used a combination of data sources, mainly observations with women surveys. Eight studies did not define the data source.

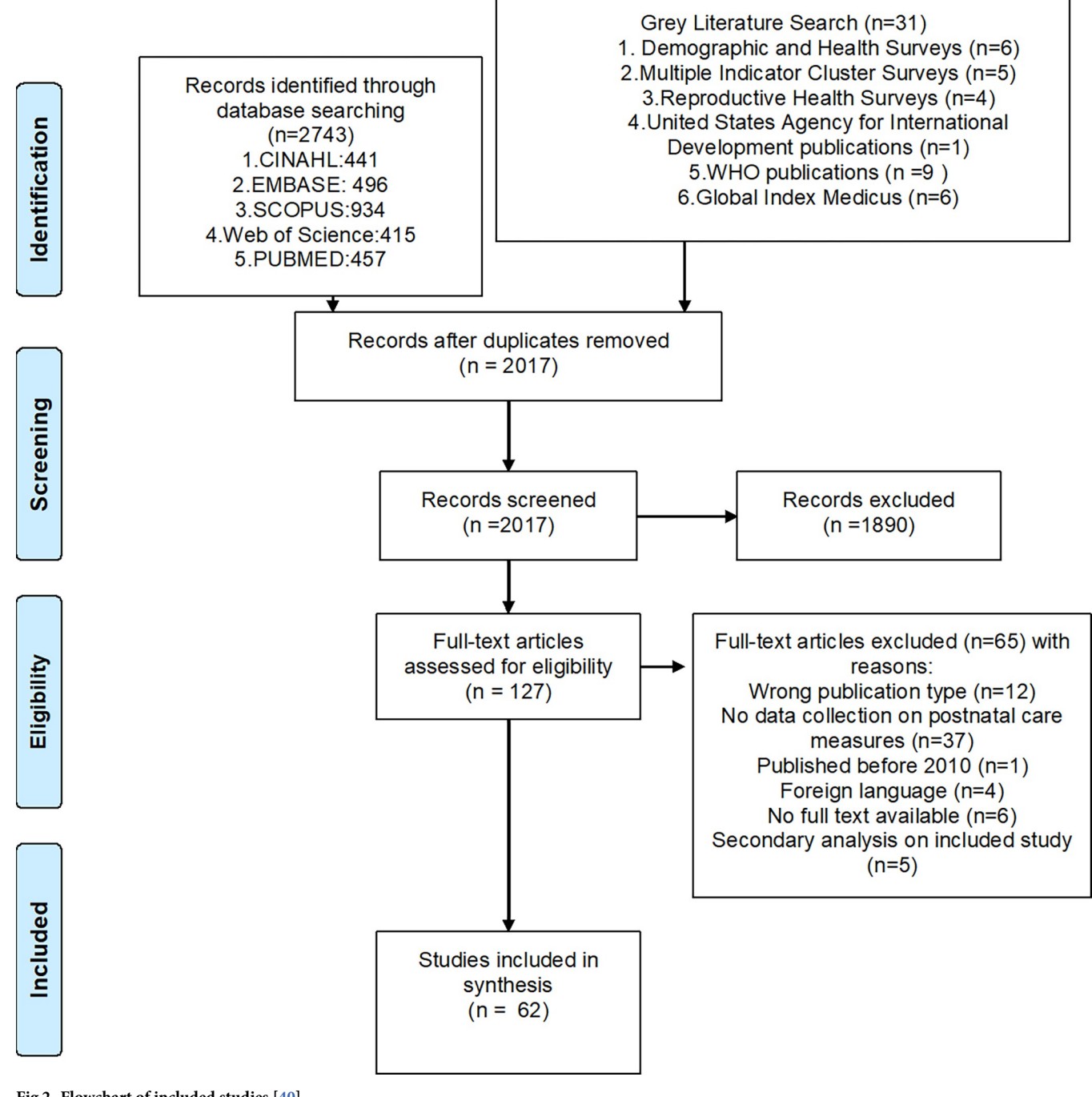

**Fig 2. Flowchart of included studies [40].**

## Provision of care measures - women

We found existing quality of care measures for 19 of the 24 numbered recommendations of the WHO PNC guideline (Table 2). Number of studies (including scientific studies and reports from grey literature) per recommendation varied from 0-18. Most studies with measures were found for the recommendation regarding postpartum contraception and physical assessment of the woman. Other recommendations with at least four different studies providing measures

**Table 1. Characteristics of the included studies.**

| WHO Region* | Number of studies in % (n) |
| --- | --- |
| African Region | 32 (n=20) |
| Region of the Americas | 8 (n=5) |
| South-East Region | 16 (n=10) |
| European Region | 16 (n=10) |
| Western Pacific Region | 8 (n=5) |
| Eastern Mediterranean Region | 2 (n=1) |
| Multi region | 18 (n=11) |
| **Study design** | |
| Cross-sectional | 61 (n=38) |
| (Quasi) experimental | 8 (n=5) |
| Guideline document | 8 (n=5) |
| Validation study | 10 (n=6) |
| Scale or instrument development | 13 (n=8) |
| **Clearly defined measures** | |
| Yes | 90 (n=56) |
| No | 10 (n=6) |
| **Data source(s)** | |
| Combination of sources | 21 (n=13) |
| Population based household data | 18 (n=11) |
| Health facility records | 7 (n=4) |
| Not defined/applicable | 13 (n=8) |
| Women survey | 31 (n=19) |
| Observations | 11 (n=7) |

* https://www.who.int/about/who-we-are/regional-offices.

were: HIV catch-up testing, local cooling for perineal pain relief, pharmacological relief of pain due to uterine cramping/involution and undertaking physical activity. The following recommendations for the care of women had no measures: limiting sedentary behavior, screening for tuberculosis disease, postnatal pelvic floor muscle training and preventive schistosomiasis treatment. Most of the recommendations with no measures are either those that are not routinely recommended or those that were context specific (Table 2). Looking at the different categories of interventions, especially measures on mental health interventions were scarce (Table 2).

## Provision of care measures – Newborns

We found existing quality of care measures for 16 of 19 numbered recommendations (Table 3). Number of studies per recommendation varied from 0-34. The following recommendations had a wide range of studies (15 or more) providing measures: assessment of the newborn for danger signs, umbilical cord care, immunization for the prevention of infections and exclusive breastfeeding. No measures were identified related to nutritional interventions (Vitamin A and D supplementation). Also, very few studies (n=2) provided measures for universal screening of the newborn (for abnormalities of the eye, hearing impairment or hyperbilirubinemia).

## Health systems and health promotion interventions

We found existing quality of care measures for 8 of 12 recommendations (Table 4). Number of studies per recommendation varied from 0-33. A high number of studies provided measures

**Table 2. Identified studies with quality of care measures on provision of maternal care as per the 2022 WHO PNC recommendations.**

| Number | Maternal assessment | References | Number of studies |
|---|---|---|---|
| 1 | Physiological assessment of the woman | [25,41–57] | 18 |
| 2 | HIV catch-up testing** | [44,52,58,59] | 4 |
| 3 | Screening for tuberculosis disease** | - | - |
| | **Interventions for common physiological signs and symptoms** | | |
| 4 | Local cooling for perineal pain relief | [45] | 1 |
| 5 | Oral analgesia for perineal pain relief | [45,52,55,60] | 4 |
| 6 | Pharmacological relief of pain due to uterine cramping/involution | [41,45,52,60] | 4 |
| 7 | Postnatal pelvic floor muscle training for pelvic floor strengthening* | - | - |
| 8 | Non-pharmacological interventions to treat postpartum breast engorgement* | [41,45] | 2 |
| 9 | Pharmacological interventions to treat postpartum breast engorgement | [45] | 1 |
| | **Preventive measures** | | |
| 10 | Non-pharmacological interventions to prevent postpartum mastitis | [41,45] | 2 |
| 11 | Pharmacological interventions to prevent postpartum mastitis | [45] | 1 |
| 12 | Dietary advice and information on factors associated with constipation | [41,45,61] | 3 |
| 13 | Routine use of laxatives for the prevention of postpartum constipation*. | [45] | 1 |
| 14 | Prevention of maternal peripartum infection after uncomplicated vaginal birth | [41,45] | 2 |
| 15 | Preventive anthelminthic treatment** | - | - |
| 16 | Preventive schistosomiasis treatment** | - | - |
| 17 | Oral pre-exposure prophylaxis for HIV prevention** | - | - |
| | **Mental health interventions** | | |
| 18 | Screening for postpartum depression and anxiety | [49,62] | 2 |
| 19 | Prevention of postpartum depression and anxiety | [62,63] | 2 |
| | **Nutritional interventions and physical activity** | | |
| 20 | Postpartum oral iron and folate supplementation | [44,49,64] | 3 |
| 21 | Postpartum vitamin A supplementation* | [49,50] | 2 |
| 22 | Physical activity | [41,45,53,62] | 4 |
| 23 | Sedentary behaviour | - | - |
| | **Contraception** | | |
| 24 | Postpartum contraception | [15,25,41,45,47,49,51,52,57,58,61,62,64–69] | 18 |

*Not routinely recommended **Context-specific recommendation.

related to the schedules for PNC contacts (Table 4), in both the peer-reviewed and grey literature. All other recommendations regarding the health system and health promotion interventions had four or less studies providing measures. Specific quality of care measures focusing on home-based care (recommendation 48 and 53) were limited. For the recommendations regarding the use of digital technologies for targeted communication and using digital birth notifications, no measures were found in the literature.

## Experience of care measures

We found experience of care measures for i) maternal care, ii) newborn care and iii) health system and health promotion interventions (Table 5). The highest number of studies was found for measures regarding the experience of maternal care. Experience of maternal care measures

**Table 3. Identified studies with quality of care measures on provision of newborn care as per the 2022 WHO PNC recommendations.**

| Number | Newborn assessment | References | Number of studies |
|---|---|---|---|
| 25 | Assessment of the newborn for danger signs | [14,15,25,41–45,48–51,53–55,57–59,61,66,70–74] | 25 |
| 26 | Universal screening for abnormalities of the eye | [45] | 1 |
| 27 | Universal screening for hearing impairment | [45] | 1 |
| 28 and 29 | Universal screening for neonatal hyperbilirubinemia | [42] | 1 |
| | **Preventive measures** | | |
| 30 | Timing of first bath to prevent hypothermia and its sequelae | [49,52,70–72,75–77] | 8 |
| 31 | Use of emollients for the prevention of skin conditions* | - | - |
| 32 | Umbilical cord care | [14,15,44,49,52,57,70,71,76–82] | 15 |
| 33 | Sleeping position for the prevention of sudden infant death syndrome | [62] | 1 |
| 34 | Immunization for the prevention of infections | [25,41,44,45,47–53,57,58,64,75,76,83,84] | 18 |
| | **Nutrition Intervention** | | |
| 35 | Neonatal vitamin A supplementation** | - | - |
| 36 | Vitamin D supplementation for breastfed, term infants** | - | - |
| | Infant growth and development | | |
| 37 | Whole-body massage | [62] | 1 |
| 38 | Responsive care to infants and children 0 and 3 years of age | [45,52,60,62–64,67,79] | 8 |
| 39 | Early learning activities for infants and children 0 and 3 years of age | [45] | - |
| 40 | Integration of responsive care and early learning and nutrition interventions | [45,51,59,62,66] | 5 |
| 41 | Integration of interventions to support maternal mental health into early childhood services | [62] | 1 |
| | **Breastfeeding** | | |
| 42 | Exclusive breastfeeding | [14,25,41,43,44,48–50,52,53,55–58,63,64,68–71,75–82,84–89] | 33 |
| 43a | Health-facility written breastfeeding policy communicated to staff and parents. | [57] | 1 |
| 43b | Health-facility staff knowledge, competence and skills to support women to breastfeed. | [25,45,51,57,63,67,79] | 7 |

*Not routinely recommended **Context-specific recommendation.

often focused on specific concepts such as satisfaction [51,53,62,66,67,79,86,93] or respectful care [15,25,41,91,94,95] in a clinical setting. Studies including experience of care measures related to newborn care included satisfaction with (information related to) newborn care [53,60,63,91,96] or if contact with the newborn was facilitated [95]. Studies concerning experience of care measures related to health system and health promotion interventions included measures related to continuity of care [45,60,62,63,93], involvement or support of the woman's partner [60,62,91,93,95] and home-visits [60,67,86]. All studies measuring experience of postnatal care considered the perspective of the woman only and collected data from women only.

## Validity of quality of care measures

In 24 out of 62 studies (39%) the authors conducted a validity assessment of the used quality of care measure. Methods used in the different studies to validate indicators or measures varied

**Table 4. Identified studies with quality of care measures for health systems and health promotion interventions as per the 2022 WHO PNC recommendations.**

| Number | Health systems and health promotion interventions | References | Number of studies |
|---|---|---|---|
| 44 | Schedules for postnatal care contacts | [14,41,45,47,49,51,52,62,64,68,69,71,74,75,80,83,84,87,89,90] | 20 |
| 45 | Length of stay in health facilities after birth | [57,67] | 2 |
| 46 | Criteria to assess prior to discharge from the health facility after birth | - | - |
| 47 | Discharge preparation/readiness approaches | [43,45,57,62] | 4 |
| 48 | Home visits for postnatal care contacts | [45,64,74] | 3 |
| 49 | Midwifery continuity of care* | [43,60,62,67] | 4 |
| 50 | Task sharing components of postnatal care service delivery | [45,62] | 2 |
| 51 | Recruitment and retention of staff in rural and remote areas | - | - |
| 52 | Men's Involvement in postnatal care * | [62,91,92] | 3 |
| 53 | Home-based records | [45] | 1 |
| 54 | Digital targeted client communication** | - | - |
| 55 | Digital birth notifications** | - | - |

* Recommended with targeted monitoring and evaluation **Context-specific recommendation.

(Fig 3). Of the 24 studies reporting validation, seven used Cronbach alpha to assess for reliability [41,60,63,67,96–98], eight studies used area under the receiver operating characteristic curve (AUC) [47,48,52,55,56,58,88,93] and two studies used confirmatory factor analysis (CFA) [53,96]. One study used the Delphi method, and one used an external criterion [99]. Three studies used both Cronbach alpha and CFA [46,91,100]. For three studies it was unclear what the author used to validate the given quality of care measure [25,94,95].

## Measures not mapped to the WHO PNC recommendations

PNC quality of care measures not mapped to the recommendations can be found in a supplementary file (S1 Table). These included quality of care measures for care immediately after childbirth, which is not addressed in the 2022 WHO PNC guideline but in the WHO guideline "Intrapartum care for a positive childbirth experience". Also specific quality of care measures for small or sick newborns could not be mapped to the recommendations, since the guideline is targeting routine care for healthy newborns and women after childbirth. Other measures were often very (context) specific (e.g. "did you receive a hot drink"), making it impossible to map them to 2022 WHO PNC recommendations.

## Gaps in measurement

Research gaps were identified for three of the four listed categories of the framework of Robinson et al. (2011) [39]. We found 11 recommendations [recommendation 3,7,15-

**Table 5. Identified studies with quality of care measures for experience of care as per 2022 WHO PNC recommendations.**

| EXPERIENCE OF CARE | References | Number of studies |
|---|---|---|
| Maternal Care | [15,25,41,45,46,52,53,59,60,62,63,86,91,93–97] | 18 |
| Newborn Care | [53,60,63,91,95,96] | 6 |
| Health system and health promotion interventions | [45,60,62,63,86,91,93,95] | 8 |

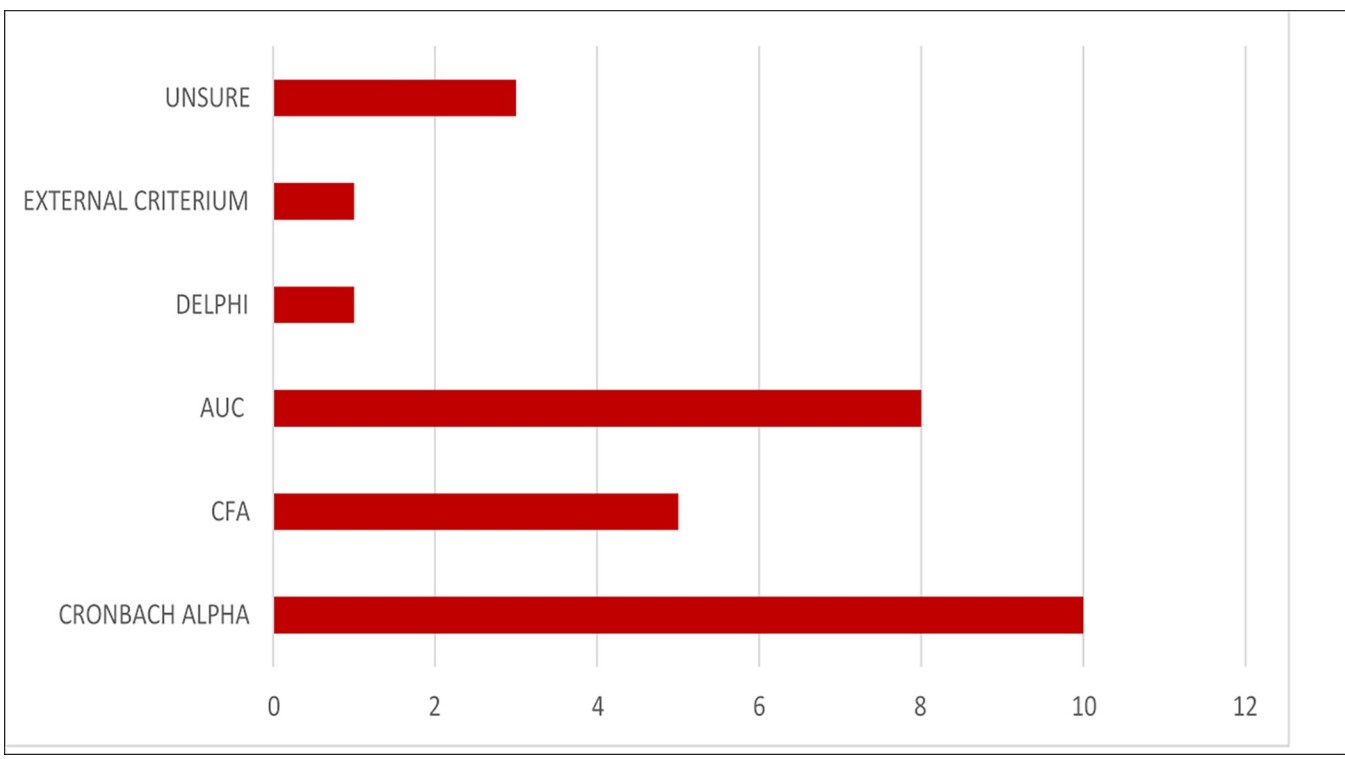

**Fig 3. Different methods used to validate quality of care measures within the studies.** Legend: AUC=Area Under the receiver operating characteristic Curve; CFA=Confirmatory Factor Analysis.

17,23,35,36,46,54,55] with no available quality of care measures in the literature (category 1: insufficient or imprecise information). In addition, we observed that all experience of care data were collected from women only, while the 2022 WHO PNC guideline recommends responsive care for women, their partners and families. Only one study also asked questions about how the partner was treated as an element of experience of care [91]. Majority of studies also focused on very specific elements of PNC such as the care provided in the facility within 48 hours, women's satisfaction with care or the interventions performed during a postnatal consultation at the facility. Studies providing PNC measures combining different aspects of PNC quality (provision of and experience of care) included the postnatal care quality standards by the National Institute for Health and Care Excellence (NICE) [62], the multidimensional satisfaction questionnaire (the WOMBPNSQ) developed by Smith (2011) [93], quality indicators for PNC after discharge by Helsloot et al. (2011) [45] and previously developed quality standards by WHO [25,44,57].

Examining gaps in the literature according to the second category (category 2: biased information), we found that 61% (n=38) of the studies did not report any validation or testing of measures, which means these studies might use measures with low validity. Studies reporting validated quality of care measures often provided little information on how this validity assessment was performed.

Lastly, we found that the definition/operationalization of the quality of care measures per recommendation varied widely in the different studies (category 3: inconsistent results). The recommendation regarding a physical assessment of the women for example included a wide variety of measures. Some measures included different components of a physical examination such as blood pressure screening, abdominal exam, check of anemia, while others did not

specify subcomponents or used a different set of physical examinations. Definitions of most measures (e.g., measures related to breastfeeding counselling or receiving a PNC check-up) also varied according to the timeframe they used: within one hour, within 24h, within two days and within 6 weeks. This implies that some wide used measures (such as exclusive breastfeeding or family planning counselling) were operationalized differently within different study settings and as a consequence cannot be compared. Some proposed measures were also poorly defined in the articles or very broadly defined. Such broadly defined measures included "adequate post-natal counselling", "treated with respect at all times", or "receiving breastfeeding counselling".

For the last category of the framework ("Not the right information") no specific research gaps were identified.

## Discussion

In this review we gave an overview of existing measures assessing quality of PNC mapped to the 2022 WHO PNC recommendations. We found that for a small number of recommendations related to broader global health issues e.g., breastfeeding and postpartum contraception an array of measures existed while for recommendations addressing more specific postnatal care issues e.g., nutritional interventions, universal screening of the newborn, mental health interventions and home-based care measures were scarce. Also, for relatively new recommendations such as on sedentary behavior and criteria to be assessed before discharge no measures existed.

Several studies within our review used a limited set of measures (e.g., the percentage of women and newborn with a postnatal check-up within two days or the percentage of women being satisfied with care) to assess PNC "quality". Some measures were also poorly defined, leaving doubt on what care was actually provided. For this reason, "uptake of PNC" was not considered as a measure for quality of postnatal care within our review, because uptake alone does not reveal actual provision (what care was provided) or experience of care as outlined in the WHO quality of care framework [8]. Further efforts among researchers to be transparent and distinguish the different concepts of uptake, coverage, content and the umbrella term "quality of care" are needed to improve comparability of data across different studies and settings. Only a minority of studies provided measures for a broader understanding of quality of care, including both provision of and experience of care [37]. Recognition of the multifaceted nature of quality of care is critical for prioritising health interventions within postnatal care and increasing uptake of services along the continuum [101].

The tendency to focus on individual clinical interventions (and corresponding measures) for the woman or the newborn might be related to the fragmented organization of PNC in the health system [102,103]. In almost all countries, PNC is still strictly divided into women and newborn services, often from different health providers, with services sometimes further sub-divided into different specialties such as nutrition, family planning and neonatal screening [102,104,105]. This approach towards PNC fails to recognize the mother-baby dyad and might create barriers in accessing a comprehensive package of care [3,4,6,104,105]. While globally the importance of integrated PNC has been recognised, policies and practise are still lagging behind [45,104,106]. Together with a change in organisation of PNC on the ground, measures to evaluate quality of PNC should also aim for a more holistic assessment. Ideally, quality of PNC is measured by considering both provision and experience of care and also recognizing the importance of the mother-baby dyad.

Noteworthy, experience of care measures within our review all relied on data from women only. Also including the perspective of partners, parents, caregivers and families will be needed to guarantee a positive postnatal care experience for all people involved.

Majority of measures were focused on facility-based care, revealing a gap in measures covering the postnatal period after discharge at home. Over the last years many countries have shortened the length of stay after birth [107–109], with home visits by trained health workers becoming a more important part of PNC [110,111]. More quality of care measures focusing on the care at home (including professional care by health care workers, self-care and family care practices in the home) will be needed to also monitor and guarantee PNC quality outside the facility.

There were inconsistencies in how PNC quality measures were defined and measured in different studies and data collection platforms, hampering comparability across settings. Two recent reviews on antenatal care measures and measures on the experience of facility-based care reported the same challenges [21,22], showing that the lack of consensus on well-defined measures seem to be an overarching problem in the field maternal and newborn health. In order to improve comparability across settings, clearly defined PNC quality measures need to be prioritized together with standardized meta-data for those indicators.

In line with other reviews on quality of care measures [21,112], we found that most indicators were not validated or tested. Indicator validation and testing is complicated, as there are various methods depending on the type of indicator and type of data collection platform. Recently "The Improving Coverage Measurement (ICM) Core Group" has developed a standard for population-based intervention coverage indicators, but this is a very time and resource intensive procedure not suitable for all quality of care indicators [113]. WHO and other researchers rather propose that indicator validity should be considered as an ongoing process without rigid cut-off points [114,115]. In order to facilitate harmonized monitoring and evaluation of maternal and newborn health measures, WHO has developed an online toolkit with guidance for indicator validation and testing [114]. After defining priority indicators for monitoring the quality of PNC, additional validation and testing work will be needed to ensure robust measures within different data collection platforms.

This review has some methodological limitations. The keywords "postnatal care", "quality of care" and "measures" were used in our search strategy. However, some articles might have included important measures for PNC quality but might not have referred to these keywords. Especially studies on experience of care might not refer to "measures" or "indicators", but still provide important data on experience of care measurement. An additional review specifically focusing on experience of care in the postnatal period might be needed to identify all available experience of care measures within the literature. Strengths include a rigorous process by adhering to the PRISMA-ScR guidelines for scoping reviews and screening the literature by two researchers independently [28]. Lastly, possible publication bias was minimised by incorporating a wide range of non-peer-reviewed papers and (survey) reports.

## Conclusion

The 2022 WHO PNC guideline provides comprehensive recommendations for strengthening postnatal care for a positive postnatal experience for women and newborns. However, currently measures are inadequate in distribution and standardisation for effective monitoring quality PNC as depicted in the 2022 WHO PNC guideline. The development of a monitoring framework will require reviewing the quality of care measures identified in this review, a process for prioritization, as well as the development of a research agenda to develop and test new measures as appropriate that can robustly capture the quality of PNC.

## Supporting information

**S1 Checklist. Meta-Analyses extension for Scoping Reviews (PRISMA-ScR) checklist.** (PDF)

**S1 Table. Data extraction sheet.**
(XLSX)

**S1 Text. Search strategies.**
(PDF)

## Acknowledgments

We would like to thank the librarian of Ghent University, Ms. Nele Pauwels, for reviewing and improving the search strategy.

## Author Contributions

**Conceptualization:** Anna Galle, Allisyn C. Moran, Ann-Beth Moller.

**Data curation:** Anna Galle, Katriona Graham, Godwin Kwaku Tuabu, Bianca De Sá é Silva, Ann-Beth Moller.

**Formal analysis:** Anna Galle, Katriona Graham, Ann-Beth Moller.

**Methodology:** Anna Galle, Ann-Beth Moller.

**Project administration:** Allisyn C. Moran.

**Supervision:** Allisyn C. Moran, Ann-Beth Moller.

**Writing – original draft:** Anna Galle.

**Writing – review & editing:** Allisyn C. Moran, Mercedes Bonet, Moise Muzigaba, Anayda Portela, Louise Tina Day, Ann-Beth Moller.

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
