## [Decision Letter · Decision Letter 0]

23 Dec 2022

PGPH-D-22-01844

Measures to assess quality of postnatal care: a scoping review

Dear Dr. University,

Thank you for submitting your manuscript to PLOS Global Public Health. After careful consideration, we feel that this paper can be accepted after addressing the minor comments from the reviewer. Therefore, we invite you to submit a revised version of the manuscript that addresses the points raised during the review process.

We look forward to receiving your revised manuscript.

Kind regards,

Stephen J. McCall, DPhil

Academic Editor

Journal Requirements:

Additional Editor Comments (if provided):

Thank you for this well written and timely paper.

Reviewers' comments:

Reviewer's Responses to Questions

**Comments to the Author**

1. Does this manuscript meet PLOS Global Public Health’s publication criteria? Is the manuscript technically sound, and do the data support the conclusions? The manuscript must describe methodologically and ethically rigorous research with conclusions that are appropriately drawn based on the data presented.

Reviewer #1: Yes

2. Has the statistical analysis been performed appropriately and rigorously?

Reviewer #1: N/A

3. Have the authors made all data underlying the findings in their manuscript fully available (please refer to the Data Availability Statement at the start of the manuscript PDF file)?

Reviewer #1: Yes

4. Is the manuscript presented in an intelligible fashion and written in standard English?

Reviewer #1: Yes

5. Review Comments to the Author

Reviewer #1: Thank you for the opportunity to review this excellent manuscript. It is well written, timely,informative and the methods are well documented. I only have minor comments, mostly related to the phrasing/naming of key concepts.

1. The authors refer to the 2022 WHO PNC recommendations, but they also use the term “PNC guideline” to refer to this document. Is this a correct and accepted synonym? They also use the word “recommendation” to mean an individual element of care which is listed in the document, so maybe you decided to use the word “guideline” to differentiate these two concepts? Please can you explain in 1 sentence? The document is already heavy on terms and clarity can be improved.

2. At the end of the Introduction section (as well as in the title), the focus of the study is on “quality of postnatal care”. I was really excited about this clear focus, and thought that it would be clearly differentiated from coverage. However, there was no information provided in the Introduction about how the authors defined “quality”. This would be really helpful because many (including me) struggle when reviewing literature to separate various concepts of coverage, content, contact and quality – it would be helpful to the reader to follow how the authors approached this given they are looking at a fairly large body of literature. It might also help to refer here to widely accepted frameworks, such as the WHO maternal healthcare quality framework – it can be helpful for synthesis in Results and Discussion. It is mentioned on page 7, but only in terms of helping identify search terms, not as an analytical framework or a way to help synthesise findings and identify gaps

3. To continue with the point above, in the Methods section -> Inclusion criteria, the key one is related to “Decribing quality of PNC, including provision and experience of care”. This is quite broad and fairly generic. I am none the wiser about how the author team made the “call” about including or excluding papers. What is a “quality of PNC provision” indicator? Can you give examples? For example, page 18 line 287 – you talk about percentage of women with PNC check within 2 days – this sounds to me like a measure of coverage, not a measure of quality. How did you draw the lines? Especially as “PNC uptake” alone was specifically excluded. Easiest would be to link to the WHO QoC framework if this were to be included in the Introduction, but this is just a suggestion. Did you as an author team have problems/challenges with differentiating uptake indicators from quality of provision indicators? If so, would you mind sharing this briefly with the readers (in the Discussion section, for example). This can be very insightful.

4. Going further on the point of quality, once the inclusion/exclusion criteria were explained, the manuscript hardly ever refers back to the key focus of the paper on “quality of PNC” – it uses much a more generic term “measures”. I would strongly urge the authors to keep repeating “PNC quality of care measures” to help readers stay focused on this point. Otherwise, it becomes very easy to drift off and think of all “PNC indicators” or “PNC measures”. One example is the title of Table 2 – the word quality needs to be included. Thank you for providing your clear explanation for the choice of the word measure as opposed to indicator.

5. Figure 1. The word recommendation is misspelled. You clearly show the structure of the PNC recommendations. I would like to suggest that you add the two dimensions along which you are reporting the findings: provision and experience, in this visual, for ease of orientation in the Results.

6. Page 11, line 184. Suddenly the term “resource(s)” starts to be used here. What does this mean? Do you mean it in terms of “an included document” (whether a published paper or a report)? Or is it a synonym for “measures”? I was confused in this paragraph, for example: “Number of resources per recommendation….” And then “the following recommendations for the care of women had no measures”. Are you matching the various measures to each PNC recommendation or the number of documents? Or is a resource a synonym for measure? Can one resource have multiple measures (in a sense that one paper can propose more than one indicator)? Please also look at last column in Table 2 – should this be number of resources or number of measures?

7. As a reader, I would have welcomed more concrete examples of the measures you identified in the main text, because until the Discussion section (line 287), there are no examples of the actual measures extracted from the included papers, all we are seeing is the numbers of measures which are mapping onto the various recommended elements of care. I know all this is in the Supplementary file 3. This is a discretionary comment for the authors to consider.

8. Were there measures of PNC quality that you found which did not map onto any of the PNC recommendations/elements of care? Maybe I missed this, but would be good to see.

Minor points:

1. Page 10, line 168: 32% of studies are not “most”, this means more than half. Suggest rephrase to “most common”

2. Page 7, line 135. Funding resources – I think you mean funding sources?

3. Page 18, line 297 – lacking behind or lagging behind?

4. In Supplementary material 3, you use “indicator” not “measure”, this is inconsistent with the main text.

Lenka Benova

6. PLOS authors have the option to publish the peer review history of their article (what does this mean?). If published, this will include your full peer review and any attached files.

**Do you want your identity to be public for this peer review?** For information about this choice, including consent withdrawal, please see our Privacy Policy.

Reviewer #1: No

---

## [Editor Report · Decision Letter 1]

30 Jan 2023

Measures to assess quality of postnatal care: a scoping review

PGPH-D-22-01844R1

Dear Dr. University,

We are pleased to inform you that your manuscript 'Measures to assess quality of postnatal care: a scoping review' has been provisionally accepted for publication in PLOS Global Public Health.

Best regards,

Stephen J. McCall, DPhil

Academic Editor